# Activation of the Integrated Stress Response and ER Stress Protect from Fluorizoline-Induced Apoptosis in HEK293T and U2OS Cell Lines

**DOI:** 10.3390/ijms22116117

**Published:** 2021-06-06

**Authors:** José Saura-Esteller, Ismael Sánchez-Vera, Sonia Núñez-Vázquez, Ana M. Cosialls, Pau Gama-Pérez, Gauri Bhosale, Lorena Mendive-Tapia, Rodolfo Lavilla, Gabriel Pons, Pablo M. Garcia-Roves, Michael R. Duchen, Daniel Iglesias-Serret, Joan Gil

**Affiliations:** 1Departament de Ciències Fisiològiques, Facultat de Medicina i Ciències de la Salut, Universitat de Barcelona-IDIBELL (Institut d’Investigació Biomèdica de Bellvitge), L’Hospitalet de Llobregat, 08907 Barcelona, Spain; jsauraes@ub.edu (J.S.-E.); isanchezvera@ub.edu (I.S.-V.); nunezvazquez.sonia@gmail.com (S.N.-V.); anamcosialls@ub.edu (A.M.C.); p.gamaperez@gmail.com (P.G.-P.); gpons@ub.edu (G.P.); pgarciaroves@ub.edu (P.M.G.-R.); daniel.iglesias@umedicina.cat (D.I.-S.); 2Department of Cell and Developmental Biology, University College London, London WC1E6BT, UK; g.bhosale.12@ucl.ac.uk (G.B.); m.duchen@ucl.ac.uk (M.R.D.); 3Laboratory of Medical Chemistry, Faculty of Pharmacy and Food Sciences and Institute of Biomedicine (IBUB), University of Barcelona, 08028 Barcelona, Spain; lorena.mendive@gmail.com (L.M.-T.); rlavilla@ub.edu (R.L.); 4Facultat de Medicina, Universitat de Vic-Universitat Central de Catalunya (UVic- UCC), Vic, 08500 Barcelona, Spain

**Keywords:** fluorizoline, prohibitin, apoptosis, ISR, ER stress

## Abstract

The prohibitin (PHB)-binding compound fluorizoline as well as PHB-downregulation activate the integrated stress response (ISR) in HEK293T and U2OS human cell lines. This activation is denoted by phosphorylation of eIF2α and increases in ATF4, ATF3, and CHOP protein levels. The blockage of the activation of the ISR by overexpression of GRP78, as well as an increase in IRE1 activity, indicate the presence of ER stress after fluorizoline treatment. The inhibition of the ER stress response in HEK293T and U2OS led to increased sensitivity to fluorizoline-induced apoptosis, indicating a pro-survival role of this pathway after fluorizoline treatment in these cell lines. Fluorizoline induced an increase in calcium concentration in the cytosol and the mitochondria. Finally, two different calcium chelators reduced fluorizoline-induced apoptosis in U2OS cells. Thus, we have found that fluorizoline causes increased ER stress and activation of the integrated stress response, which in HEK293T and U2OS cells are protective against fluorizoline-induced apoptosis.

## 1. Introduction

Fluorizoline is a synthetic molecule that presents a fluorinated thiazoline scaffold and induces apoptosis in cancer cell lines and primary cells, even in those that present p53 mutations [1,2,3]. Fluorizoline specifically binds to Prohibitin 1 (PHB1) and PHB2 (PHBs) [1], which regulate multiple processes within the cell, including cell signaling, gene transcription, or cell cycle progression [4,5,6,7,8]. Prohibitins are conserved proteins that locate to the mitochondria, the plasma membrane, and the nucleus. The most studied functions of PHBs take place in the mitochondria, where these proteins form large heteromultimeric ring-like complexes [9,10]. Mitochondrial PHB complexes participate in the quality control of mitochondria and regulate mitochondrial fusion and fission. Furthermore, PHBs are important in the regulation of cristae morphology, reactive oxygen species formation, or mitophagy [4,5,11]. Different molecules have been described to interact with PHBs [12,13,14]. We previously demonstrated that PHBs are required for the proapoptotic effect of fluorizoline in mouse embryonic fibroblasts as well as in HEK293T and U2OS human cell lines [15,16].

Mitochondrial stress induced by different stimuli (loss of mitochondrial membrane potential, disruption of protein import to the mitochondria, OXPHOS complex destabilization, and mitochondrial translation blockage) triggers the activation of the integrated stress response (ISR) [17]. The ISR can be activated by four protein kinases: heme-regulated inhibitor (HRI), general control nonderepressible 2 kinase (GCN2), protein kinase R (PKR), and PKR-like ER kinase (PERK); that respond to different types of stress [18,19]. This pathway is characterized by the phosphorylation of eukaryotic initiation factor 2α (eIF2α), the subsequent reduction in global protein translation and increase in the translation of certain proteins, such as activating transcription factor 4 (ATF4). The ISR participates in the early protective response against stress; however, if the insult is too strong or prolonged, it can lead to the induction of apoptosis [19,20].

Previous results from our group showed that fluorizoline locates in the mitochondria, where we found that it induced mitochondrial fragmentation, long optic atrophy 1 (L-OPA1) cleavage, and cristae disruption [15]. In addition, reduction in PHB expression has been previously shown to result in increased endoplasmic reticulum (ER) stress [21], which can subsequently activate the ISR through PERK [18]. Interestingly, increases in PHB expression have been observed in concurrence with ER stress [22,23] and have been associated with resistance to ER stress in melanoma cells [24]. Recently, we have described that the induction of apoptosis after fluorizoline treatment is mediated by the transcription factors ATF3 and ATF4, which are activated by the ISR in HeLa and HAP1 cell lines [25]. Consistently, inhibition of the ISR or downregulation of ATF4 and ATF3 in these cell lines results in increased resistance to fluorizoline-induced apoptosis.

Here, we investigate how fluorizoline treatment results in the activation of the ISR and the involvement of this pathway in fluorizoline-induced apoptosis in HEK293T, U2OS, and Jurkat cells. Our results demonstrate that fluorizoline triggers ER stress and activation of the ISR, which participate in a pro-survival response to this compound in these cell lines.

## 2. Results

### 2.1. The ISR Is Induced after Fluorizoline Treatment or PHB Downregulation

We investigated whether fluorizoline and PHB downregulation caused the activation of the ISR in HEK293T, U2OS, and Jurkat cell lines. First, these cell lines were treated with fluorizoline to determine their sensitivity to this compound. As expected, fluorizoline triggered the induction of apoptosis at concentrations in the low micromolar range in these cell lines (Figure 1a and Appendix A). HEK293T, U2OS, and Jurkat cells were treated with fluorizoline for different times, and then protein expression was analyzed by Western blot. Treatment with fluorizoline resulted in a time-dependent increase in eIF2α phosphorylation in all cell lines analyzed (Figure 1b,c and Appendix A). Moreover, fluorizoline caused a marked increase in ATF4 protein levels that could be detected as soon as 1 h after fluorizoline treatment. We analyzed the effect of fluorizoline treatment in the expression of other proteins related to the ISR and observed that fluorizoline treatment resulted in increased protein expression of C/EBP homologous protein (CHOP) and ATF3 in HEK293T and U2OS cells (Figure 1d). This increase was parallel to the previously described increase in NOXA protein expression [16]. These results demonstrate that the activation of the ISR by fluorizoline treatment is conserved in different cell lines.

In addition, we tested whether another described PHB-interacting compound, rocaglamide A, resulted in a similar activation. However, we did not observe eIF2α phosphorylation or ATF4 protein increases after treatment of HEK293T and U2OS with this compound (Appendix A). Moreover, rocaglamide A reduced the expression of ATF4 in HEK293T cells and NOXA in both cell lines, which is consistent with the inhibitory effect of this compound on protein translation [26].

Next, we investigated whether PHB downregulation also increased ISR signaling in HEK293T cells and whether reduced expression of PHBs affected the increases in ATF4 and CHOP expression by fluorizoline. Cells were transfected with specific siRNAs against PHB1 and PHB2 for 72 h and then treated with either fluorizoline or with compound 2a, which shares the structure with fluorizoline but lacks fluorine residues required for its cytotoxic activity, as a negative control [1], and thapsigargin as a positive control for the induction of ATF4 and CHOP. PHB downregulation caused a marked increase in ATF4 protein levels, which was further increased after the addition of fluorizoline and thapsigargin but not in the presence of the inactive compound 2a (Figure 2). A similar effect was observed in CHOP protein expression. The expression of NOXA, instead, was basally reduced by PHB downregulation in these cell lines, but it could still be induced by fluorizoline treatment. These results indicate that PHB downregulation results in the activation of the ISR in HEK293T cells.

Recently, we showed that the induction of NOXA protein expression after fluorizoline treatment in HeLa and HAP1 cells was mediated by ATF4 and ATF3 [25]. ATF4 has been described as being critical for the induction of CHOP in different cell lines, which, in turn, participates in the induction of apoptosis by the ISR [27]. HEK293T and U2OS cells were transfected with specific siRNAs against *ATF4* and then treated with fluorizoline for up to 24 h to analyze the importance of the ATF4 protein increase in fluorizoline-induced apoptosis. ATF4 protein expression was effectively reduced in both cell lines (Appendix A). Surprisingly, and opposite to our recent report in HeLa and HAP1 cells, HEK293T cells with lower ATF4 expression had increased sensitivity to fluorizoline-induced apoptosis, while ATF4 downregulation did not affect the percentage of U2OS cells undergoing apoptosis after fluorizoline treatment (Appendix A). The cleavage of the classic apoptosis markers PARP and caspase 3 was used to confirm this result by Western blot (Appendix A).

Unexpectedly, downregulation of ATF4 in both cell lines had no effect on CHOP protein expression (Appendix A), demonstrating that CHOP is not increased by fluorizoline due to an activation of its transcription by ATF4. Given that ATF4 downregulation had no effect on the increase in CHOP protein levels after fluorizoline treatment and considering the wide association of CHOP with the induction of apoptosis downstream of ISR activation, we decided to test the role of CHOP on fluorizoline-induced apoptosis. To this end, HEK293T cells and U2OS cells were transfected with specific siRNAs against *CHOP* for 48 h and then treated with fluorizoline for up to 24 h. CHOP protein increases were completely blocked in si*CHOP* transfected cells (Appendix A). Downregulation of CHOP did not prevent the induction of apoptosis by fluorizoline in HEK293T cells, while U2OS cells with reduced CHOP protein expression were found to be more sensitive to this compound (Appendix A). This effect was again confirmed by Western blot using the apoptosis markers PARP and caspase 3 (Appendix A).

Furthermore, we used ISRIB, an established inhibitor of the ISR through the activation of the eIF2B complex [28,29], to determine whether ATF4, ATF3, CHOP, and NOXA protein increases observed after fluorizoline treatment in these cell lines were dependent on the phosphorylation of eIF2α. Consistent with the results obtained with ATF4 downregulation, ISRIB increased the sensitivity of HEK293T to fluorizoline-induced apoptosis (Appendix A). In concordance, ISRIB pre-treatment resulted in increased caspase 3 cleavage by fluorizoline (Appendix A). As expected, ISRIB prevented the induction of ATF4 and CHOP by thapsigargin. The addition of ISRIB reduced the increase in ATF4 and CHOP after 4 h of treatment with fluorizoline (Appendix A), indicating that the induction of these two proteins is caused by the phosphorylation of eIF2α. The induction of ATF3 by fluorizoline was higher after pre-treatment with ISRIB, despite ATF4 levels were partially reduced.

### 2.2. Fluorizoline Induces ER Stress in HEK293T and U2OS Cells

To determine whether fluorizoline treatment could cause ER stress, we analyzed the splicing of XBP1, which takes place after the activation of the ER sensor inositol-requiring enzyme 1 (IRE1). HEK293T and U2OS cells were treated with fluorizoline for different times, and the presence of spliced isoforms of XBP1 was assessed. We detected increased splicing of XBP1 both at the mRNA (Figure 3a) and protein levels (Figure 3b). The increase in IRE1 activity shown by XBP1 splicing suggests that the activation of the ISR after fluorizoline treatment might be caused by an increase in ER stress.

Next, we investigated the role of ER stress in fluorizoline-induced apoptosis. To this end, we overexpressed the ER luminal glucose-regulated protein 78 (GRP78) in HEK293T cells. GRP78 binds to and inhibits the ER membrane sensors PERK, IRE1, and ATF6, which are activated upon accumulation of misfolded proteins in the ER [30]. HEK293T cells were transfected with a plasmid containing human *GRP78* for 48 h and then treated with fluorizoline for 24 h. A marked increase in GRP78 protein expression could be observed by Western blot in *GRP78*-transfected cells (Figure 3c). Overexpression of GRP78 resulted in increased sensitivity to fluorizoline-induced apoptosis measured by phosphatidylserine exposure (Figure 3d) as well as increased PARP and caspase 3 cleavage (Appendix A). These cells had an attenuated activation of the ATF4, CHOP, ATF3, XBP1, and also NOXA after fluorizoline treatment (Figure 3c), indicating that ER stress is involved in the increase in these proteins.

Upon ER stress, folding capacity is impaired, and unfolded proteins are degraded mostly by the ubiquitin–proteasome system [31]. To further study the induction of ER stress by fluorizoline, we analyzed whether treatment with this compound resulted in the accumulation of poly-ubiquitinated proteins. Jurkat cells were treated either with fluorizoline or with the proteasome inhibitor MG132 as a positive control, and the presence of poly-ubiquitinated proteins was assessed by Western blot. Both MG132 and fluorizoline treatment resulted in the accumulation of poly-ubiquitinated proteins (Appendix A).

Altogether, these results demonstrate that fluorizoline causes increased ER stress and that the ER stress response is protective against fluorizoline-induced apoptosis in HEK293T and U2OS cells.

### 2.3. Fluorizoline Does Not Trigger Apoptosis through Altered Mitochondrial Respiration or ROS Production

Next, we sought to investigate how the binding of fluorizoline to PHBs in the mitochondria resulted in increased ER stress. PHB are highly conserved proteins that have been involved in numerous activities in the mitochondria and alterations of mitochondrial homeostasis, such as ATP depletion or increased reactive oxygen species (ROS) formation, can cause ER stress [32,33]. Downregulation of PHBs has been shown to increase ROS production and impair the assembly of electron transport chain complexes affecting OXPHOS activity [34,35]. Specifically, downregulation of PHBs has been described as reducing Routine, ATP coupled, and maximal respiration. We had previously used Jurkat cells to analyze ROS production and mitochondrial membrane potential after fluorizoline treatment [15]. In this cell line, fluorizoline caused increased ROS production without altering mitochondrial membrane potential [15]. To expand these results, we analyzed the production of mitochondrial superoxide after fluorizoline treatment using MitoSOX in HEK293T, U2OS, and Jurkat cells. Antimycin A was used as a positive control. Treatment with fluorizoline caused a small but significant increase in MitoSOX fluorescence in HEK293T and U2OS cells (Figure 4a,d). This effect could not be observed in Jurkat cells in which fluorizoline treatment did not alter MitoSOX fluorescence to any extent (Appendix A). We next decided to test the relevance of these increases in superoxide production for the mechanism of action of fluorizoline. First, we analyzed whether the addition of the antioxidant MnTBAP could prevent the increase in MitoSOX fluorescence produced by fluorizoline treatment. MnTBAP caused a partial reduction in the production of mitochondrial superoxide in both cell lines (Appendix A). Then, we analyzed the effect of MnTBAP on cell viability. MnTBAP itself decreased cell viability, an effect that was also observed in the presence of fluorizoline (Figure 4b,e). In addition, the induction of ATF4 after 1 h of treatment with fluorizoline was reduced in the presence of MnTBAP (Figure 4c,f). Despite this, no differences were observed after 4 h of treatment with fluorizoline. Considering that MnTBAP did not prevent neither the loss of cell viability nor the increase in ATF4 protein levels after fluorizoline treatment, we conclude that the superoxide production detected in HEK293T and U2OS cells does not participate in the induction of apoptosis by fluorizoline.

Moreover, we analyzed whether fluorizoline treatment could modulate GSH and GSSG concentration in Jurkat cells (Appendix A). Fluorizoline treatment produced no changes neither in GSH and GSSG concentration nor in GSH/GSSG ratio in Jurkat cells. These results indicate that treatment with fluorizoline does not result in increased mitochondrial superoxide production or alter the redox balance in Jurkat cells.

In order to test whether treatment with fluorizoline resembled *PHB* knockdown regarding impairment of OXPHOS activity, we measured oxygen consumption in HEK293T, U2OS, and Jurkat cells after fluorizoline treatment (Figure 5 and Appendix A). Cells were treated with fluorizoline for 4 h, and then oxygen consumption was measured. Treatment with fluorizoline resulted in a slight, non-significant reduction in oxygen consumption in all parameters measured in all cell lines (Figure 5a,c and Appendix A) which was proportional to the basal measurement (Figure 5b,d and Appendix A). Thus, fluorizoline does not modify oxygen consumption. Altogether these results indicate that fluorizoline causes an increase in superoxide production in HEK293T and U2OS cells which is not required for the induction of apoptosis as it is not conserved in all cell lines, as no alterations have been detected in Jurkat cells. Moreover, fluorizoline did not modify relative respiration rates in any of the cell lines analyzed.

### 2.4. Fluorizoline Alters Calcium Homeostasis to Trigger Apoptosis in U2OS Cells

Considering the strong and quick induction of ER stress produced by fluorizoline treatment, we hypothesized that fluorizoline might cause alterations of calcium homeostasis. Conditions that modify calcium concentration in the ER cause impaired protein folding, and therefore ER stress, resulting in the activation of ER stress-responsive pathways, such as the ISR. Interestingly, PHBs have been described as interacting with VDAC [36] channels which participate in the control of calcium flux from the ER to the mitochondria [37,38].

We investigated whether treatment with fluorizoline could result in altered cytosolic and mitochondrial calcium concentrations in human cell lines. To this end, we used the calcium-sensitive fluorescent dye Rhod-2/AM (Rhod-2), which preferentially accumulates in the mitochondria, although also in the cytosol and other organelles. As we observed a diffuse signal after Rhod-2 staining in HEK293T and U2OS cells, we decided to use MitoTracker Deep Red to discriminate mitochondrial from cytosolic Rhod-2 signal (Appendix A). Live-cell imaging was used to record variations in Rhod-2 fluorescence after treatment with either ATP as a positive control, the vehicle DMSO as a negative control, or fluorizoline (Figure 6a,d and Appendix A). In the case of HEK293T, the increase in Rhod 2 fluorescence produced by the addition of fluorizoline could not be distinguished from the addition of the vehicle DMSO (Figure 6a and Appendix A). Importantly, fluorizoline treatment caused increased Rhod-2 fluorescence in U2OS cells, starting approximately 30 s after the addition of the drug (Figure 6d and Appendix A). Rhod-2 signal increased similarly in MitoTracker Deep Red positive and negative pixels indicating that these increases in calcium concentration occurred both in the mitochondria and in the cytosol (Appendix A). Next, we analyzed whether the increase in calcium concentration was modified by PHB downregulation. Increased calcium concentration was detected after fluorizoline treatment in conditions of PHB-downregulation compared to SC (Appendix A). In addition, we transfected HEK293T cells with cytosolic and mitochondrially targeted calcium-sensitive aequorin and analyzed luminescence after fluorizoline treatment. Aequorin transfection efficiency was normalized with the use of digitonin, as described in Materials and Methods. Treatment with fluorizoline, as well as with the positive control ATP, caused an increase in luminescence both in the cytosol and in mitochondria (Appendix A). Therefore, fluorizoline causes an increase in calcium concentration in these two subcellular compartments in both U2OS and HEK293T cells.

To determine whether the observed calcium mobilization after fluorizoline treatment was required for fluorizoline-induced apoptosis, we analyzed the effect of BAPTA/AM (BAPTA), a cytosolic calcium chelator, in both U2OS and HEK293T cells. U2OS and HEK293T cells were pre-treated with BAPTA for 1 h and then treated with fluorizoline for 24 h. The addition of BAPTA had no effect on the induction of apoptosis by fluorizoline in HEK293T cells (Figure 6b). Importantly, pre-treatment with BAPTA reduced fluorizoline-induced apoptosis in U2OS cells in a dose-dependent manner (Figure 6e). Pre-treatment with BAPTA reduced ATF3, CHOP, and NOXA protein increases by fluorizoline in both cell lines, while ATF4 induction was only blocked in U2OS cells, being increased to a higher extent in HEK293T when BAPTA was present (Figure 6c,f). This result was confirmed using another calcium chelator, EGTA/AM (EGTA), which also prevented fluorizoline-induced apoptosis in U2OS cells, although to a lower extent (Appendix A). Thus, fluorizoline can cause rapid increases in calcium concentration that are prior to the induction of ER stress and are required for the induction of apoptosis in U2OS cells.

## 3. Discussion

In the present article, we have shown the activation of the ISR by the PHB-binding compound fluorizoline in the human cell lines U2OS, HEK293T, and Jurkat. Fluorizoline treatment increases ER stress in these cell lines, which is not caused by direct modulations of the OXPHOS system, increased ROS production, or by alterations in GSH concentration. Finally, we described an increase in calcium concentration in the mitochondria and cytosol upon fluorizoline treatment that participates in the mechanism of fluorizoline-induced apoptosis in U2OS cells.

We previously showed that NOXA, BIM, and PUMA participate in the induction of apoptosis after fluorizoline treatment, although their relative importance is cell-line specific [15,16]. These proteins have been shown to be effectors of apoptosis downstream of ER stress or to be modulated by common ER stress-responsive proteins [39,40,41,42]. For example, ATF4 and ATF3 bind to the promoter of NOXA and regulate its expression after DNA damage induced by cisplatin [43]. BIM is controlled at the transcriptional level by CHOP and at the post-translational level by JNK and ERK through phosphorylation in different residues [44,45,46,47]. In its turn, PUMA has been shown to be controlled by E2F1 downstream of ER stress [48]. The presence of ER stress and the activation of the ISR could explain how different BH3-only proteins are activated after fluorizoline treatment and might be required for the induction of apoptosis in certain cell lines. However, our results indicate that in the case of U2OS and HEK293T cells, these pathways play mainly a pro-survival role.

We observed increased activation of the integrated stress response after fluorizoline treatment, an effect that was reproduced by PHB downregulation. Of note, treatment of PHB-downregulated cells with fluorizoline further increased ATF4 and CHOP protein levels. As downregulation of PHBs was not complete, this effect could be explained by the action of fluorizoline on the remaining pool of PHBs. However, we cannot exclude the possibility of this effect being due to an off-target effect of fluorizoline.

Several reports describe the regulation of CHOP by ATF4 under stress conditions [49,50]. Despite this, fluorizoline increased CHOP expression to the same extent in conditions in which ATF4 had been downregulated by siRNA. CHOP expression is regulated by other mechanisms, including increased translation after eIF2α phosphorylation [51] or increased transcription by cleaved ATF6 [52]. Although ATF6 could play a role in the increase in CHOP after fluorizoline treatment, this accumulation was blocked by pre-treatment with ISRIB, indicating that CHOP is induced by preferential translation of its mRNA. We also observed increased ATF3 expression in the presence of ISRIB. In HeLa and HAP1 cell lines, we have found that ATF3 plays a role in fluorizoline-induced apoptosis by modulating the levels of NOXA together with ATF4 [25]. However, in HEK293T, we did not find a reduction in NOXA expression after fluorizoline treatment when a luciferase construct of NOXA promoter with a mutated CRE binding site was compared to the wild-type sequence (data not shown) suggesting that in this cell line, the regulation of NOXA is independent of ATF4 and ATF3. Moreover, NOXA plays a minor role in the induction of apoptosis by fluorizoline in this cell line [16] ATF3 has been involved in numerous publications in response to stress and the induction of apoptosis. ATF3 can participate in apoptosis through the regulation of the expression of DR5 [53]. This is particularly interesting, as we previously described that the extrinsic pathway of apoptosis can be activated in *Bax*^−/−^/*Bak*^−/−^ MEF after fluorizoline treatment, but no effect could be observed in WT cells [15].

The finding of increased activation of IRE1 was shown by splicing of XBP1 together with the prevention of the increase in ATF4, ATF3, CHOP, and XBP1 itself by overexpression of GRP78 indicate that fluorizoline treatment results in increased ER stress. However, whether ER stress is the main cause of the activation of the ISR remains to be determined. Two recent studies by Guo et al. and Fessler et al. described the molecular pathway transmitting the signal of mitochondrial stress into the activation of the ISR [54,55]. This pathway involves the cleavage of the mitochondrial protein DELE1 by the stress-activated protease OMA1. Cleaved DELE1 accumulates in the cytoplasm, where it can facilitate the activation of the eIF2α kinase HRI and, therefore, of the ISR. OMA1 localizes in the inner mitochondrial membrane and is in charge of the cleavage of OPA1 leading to mitochondrial fragmentation and cristae junction opening [56]. Interestingly, PHBs increase the turnover of OMA1 through a mechanism possibly involving cardiolipin [57]. We have observed that treatment with fluorizoline and downregulation of PHBs in cell lines from human origin results in the activation of the ISR. It would be interesting to analyze the involvement of DELE1, OMA1, and HRI in this process as it could help to understand the mechanism of action of fluorizoline.

We observed a small increase in MitoSOX fluorescence after treatment with fluorizoline in HEK293T and U2OS cells. However, our results indicate that this alteration is not required for the induction of apoptosis in these cell lines. This is in line with previous findings of our group using MEF cells which were not protected from apoptosis by fluorizoline even when ROS production was completely blocked [15]. Fluorizoline did not increase mitochondrial superoxide production in Jurkat cells, as opposed to what has been previously described for PHB knockdown [35]. On the contrary, fluorizoline increased ROS production in Jurkat cells measured with CellROX Deep Red reagent [15]. This discrepancy might be explained by the compartment where each probe accumulates, as MitoSOX senses mitochondrial superoxide and CellROX Deep Red accumulates in the cytoplasm, which might indicate an increase in extra-mitochondrial ROS production. PHBs present mainly a structural function. The fact that fluorizoline does not mimic PHB downregulation regarding ROS production suggests that the binding of fluorizoline does not result in complete blockage of the function of PHBs, as these proteins are still expressed in the presence of fluorizoline and might maintain part of their activity or protein interactions.

We have found increases in cytosolic and mitochondrial calcium concentration after fluorizoline treatment. Fluorizoline accumulates in the mitochondria [1], which is the predominant location of PHB in most cell lines [9]. Mitochondria and ER are highly interconnected organelles, and mitochondrial associated membranes (MAMs) are an emerging field of study [58], as they control the transfer of calcium from the ER to the mitochondria, which is key to modulating mitochondrial metabolism [59]. Some of the proteins involved in the regulation of calcium transport between the ER and the mitochondria are known interactors of PHBs, such as VDACs [36,37]. In addition, PHBs in the plasma membrane are required for the activation of phospholipase-C gamma (PLCγ) downstream of CD86 and have been shown to interact with phosphatidylinositol-3,4,5-trisphosphate (PIP_3_) and, thus, may play a role in calcium signaling [60,61]. However, as fluorizoline does not accumulate in the plasma membrane, we consider it unlikely that the activity of this compound relies on an effect over PHBs present in the plasma membrane. How the interaction of fluorizoline with PHBs results in calcium mobilization from the ER remains to be explored.

Rhod-2 failed to detect calcium increases after fluorizoline treatment in HEK293T cells. In contrast, the assays with aequorin detected increased luminescence of this protein both in the cytosol and in the mitochondria. Differences in the measurement of calcium with calcium-sensitive dyes and luminescent proteins have been previously reviewed [62]. We consider that, overall, these experiments indicate that fluorizoline can alter normal calcium distribution. The addition of calcium chelators prevented the increase in proteins of the ER stress response, such as ATF3 and CHOP, in HEK293T and U2OS cells. Moreover, BAPTA pre-treatment had a heterogeneous effect on the induction of ATF4 and apoptosis, which were completely blocked by BAPTA in U2OS cells but not in HEK293T cells. The prevention of fluorizoline-induced apoptosis by EGTA in U2OS cells was lower than the one observed after pre-treatment with BAPTA. Although we cannot exclude the possibility of an unspecific effect of BAPTA-AM, this molecule has been described as presenting a much higher calcium-binding rate when compared to EGTA-AM. Thus, it is possible that the differences observed between the two calcium chelators arise from a more effective blockage of calcium concentration increase by BAPTA-AM [63]. Differences between cell lines were also found in the effect of ATF4 downregulation, which sensitized HEK293T cells but not U2OS cells to fluorizoline, and CHOP downregulation which, conversely, sensitized U2OS cells but not HEK293T cells. Thus, the relevance of this mechanism for the induction of apoptosis might depend on the cell type. Moreover, we have observed differences between cell lines regarding the effect of ISR inhibition. In HeLa and HAP1 cells, which rely on ATF4 and ATF3 to increase the expression of NOXA, blockage of the ISR resulted in increased resistance to fluorizoline-induced apoptosis [25]. Conversely, in HEK293T and U2OS cells inhibition of this signaling pathway caused increased sensitivity to fluorizoline. The molecular explanation for the different sensitivities to fluorizoline treatment between cell lines is presently unknown.

Recently, Jin X et al., considering the interaction of PHBs with proteins that participate in the regulation of translation, analyzed the effect of different PHB-binding compounds on the translation machinery [64]. These authors found a reduction in new protein synthesis upon fluorizoline treatment in human cell lines, caused by inhibition of eIF2α and eEF2. In parallel to our findings, Jin X et al. describe increased ATF4 protein levels and XBP1 splicing and point towards ER stress as the cause of these protein alterations. Even more, the authors of this study show increased calcium concentration after fluorizoline treatment in A549 cells. However, pre-treatment with ISRIB protected A549 cells to fluorizoline-induced apoptosis, similar to the results we obtained in HeLa and HAP1 cells, strengthening the idea that the activation of the ISR after fluorizoline treatment is proapoptotic or pro-survival depending on the cell type (data not shown).

In conclusion, we have found that the PHB-binding compound fluorizoline triggers ER stress and activation of the ISR in human cell lines, which participate in a pro-survival response to this compound in HEK293T and U2OS cells (Figure 7). We have observed calcium mobilization, which could explain how fluorizoline alters the ER homeostasis. Further experiments will be aimed at analyzing the mechanism leading to this increase in calcium concentration and whether it is caused by the interaction of fluorizoline to PHBs.

## 4. Materials and Methods

### 4.1. Reagents

Fluorizoline and compound 2a were synthesized as described in [1]. Thapsigargin, ISRIB, antimycin A, rocaglamide A, MnTBAP, and EGTA/AM (EGTA) were from Sigma-Aldrich (Saint Louis, MO, USA) ATP was from Roche (Basel, Switzerland). BAPTA/AM was from Abcam (Cambridge, UK). IN-8 and SB203580 were from Calbiochem (Merck-Millipore, Darmstadt, Germany).

### 4.2. Cell Lines Culture

HEK293T and U2OS cells (both from American Type Culture Collection, Rockville, MD, USA) were grown in DMEM complemented with 10% FBS, 2 mM L-glutamine, 100 U/mL penicillin, and 100 ng/mL streptomycin. Jurkat cells were maintained in RPMI 1640 complemented with 10% FBS (previously inactivated by heating at 60 °C for 30 min), 2 mM L-glutamine, 100 U/mL penicillin, and 100 ng/mL streptomycin (all from Biological Industries, Beit Haemek, Israel). All cell lines were maintained at 37 °C in a humidified atmosphere containing 5% CO_2_.

### 4.3. Transient Transfection

For RNA interference, cells were transfected using Lipofectamine RNAiMAX reagent (ThermoFisher Scientific, Waltham, MA, USA). DMEM was replaced with OptiMEM (Gibco, ThermoFisher), and complexes were added into cells dropwise and incubated for 4–6 h and then replaced again by fresh DMEM. The specific small interference RNAs (siRNAs) used were CHOP (VHS40605), PHB1 (HSS182281), and PHB2 (HSS117606) from ThermoFisher. ATF4 (UUCUCCGAACGUGUCACGU) and its respective SC (GCCUAGGUCUCUUAGAUGA) were purchased from GenePharma (Shangai, China).

For plasmid transfection, cells were transfected using Lipofectamine LTX reagent (ThermoFisher) pcDNA3.1(+)-GRP78/BiP was a gift from Richard C. Austin (Addgene plasmid # 32701; http://n2t.net/addgene:32701, accessed on 4 June 2021; RRID:Addgene_32701).

### 4.4. Western Blot

Pelleted cells were lysed with Laemmli sample buffer, and protein quantification was performed with the Micro BCA Protein Assay Reagent kit (Pierce, Rockford, IL, USA). Protein samples (15–30 µg) were reduced by the addition of DTT and then loaded into a polyacrylamide gel, and a 125-mV current was applied. For the transfer, Immobilon-P membranes (Millipore, Billerica, MA, USA) were used. Membranes were blocked with 5% (*w*/*v*) non-fat milk in Tris-buffered saline with Tween^®^ 20 and then incubated with the specific primary antibodies overnight at 4 °C. Then, primary antibodies were removed, and membranes were incubated with secondary antibodies conjugated with the horseradish peroxidase system (from GE Healthcare, Amersham, UK). Antibody binding was detected using enhanced chemiluminescence detection (GE Healthcare. The primary antibodies used were PHB1 (H-80, Santa Cruz), PHB2 (07-234, Millipore), Tubulin (CP06, Millipore), Cleaved caspase 3 (67341A, Pharmingen), β-Actin (A5441, Sigma-Aldrich), NOXA (ab13654, Abcam), p-eIF2α (Ser51) (3398, Cell Signaling), eIF2α (9722, Cell Signaling), ATF4 (sc-200, Santa Cruz), CHOP (2895, Cell Signaling), ATF3 (sc-188, Santa Cruz), XBP1s (143F, Millipore), GRP78 (3183, Cell Signaling), Ubiquitin (04-263, Millipore), and PARP (9542, Cell Signaling).

### 4.5. PCR

PCR experiments were performed with 1 μg total mRNA from samples. RNA was reverse transcribed using the High-Capacity cDNA Reverse Transcription Kit (from Applied Biosystems, Foster City, CA, USA) following the instructions from the manufacturer. cDNA was amplified using specific unlabeled primers against XBP1 (Forward: TTACGAGAGAAAACTCATGGCC; Reverse: GGGTCCAAGTTGTCCAGAATGC). The amplified DNA was separated in a gel with 8% acrylamide in 1× TAE buffer. GeneRuler DNA Ladder Mix (ThermoFisher) was used to allow for band weight determination.

### 4.6. Flow Cytometry

Phosphatidylserine exposure was used to determine cell viability. Cells were washed with PBS, pelleted, and then incubated with annexin binding buffer containing Annexin V-APC (Life Technologies) for 15 min in the dark. Then, samples were analyzed by flow cytometry using FACSCanto II and FACSDiva (BD Biosciences, Franklin Lakes, NJ, USA). Data represent the percentage of non-apoptotic cells (Annexin V-).

Mitochondrial superoxide detection was analyzed with a MitoSOX™ indicator (From ThermoFisher). Jurkat, U2OS, and HEK293T cells (0.2–0.5 × 10^6^) were collected and incubated with MitoSOX™ 15 min prior to the start of the measurement. After that, cells were washed once with 1 mL of complete PBS (c-PBS, PBS supplemented with 0.5 mM MgCl_2_, 0.7 mM CaCl_2_, and 1 g/L glucose). Fluorescence was detected by flow cytometry with FACSCanto equipment and analyzed using FlowJo software.

### 4.7. Live-Cell Imaging

Rhod-2 dye (ThermoFisher) was used to detect variations in intracellular calcium concentration. Cell medium was substituted with OptiMEM containing 5 μM Rhod-2 dye and 2 μM Pluronic acid (ThermoFisher) 30 min prior to the start of the analysis to facilitate dye loading to the cell.

Live-cell images were obtained at 1 s intervals for up to 2 min using a Carl Zeiss model LSM8880 confocal microscope. Cells were treated with 10–15 μL of the solutions adjusted to achieve the desired concentrations of the drugs 5 sec after the start of the acquisition. Quantification of the mean fluorescence intensity for each series of images was performed with ImageJ software.

### 4.8. Calcium Measurement with Aequorin

HEK293T cells were transfected with plasmids containing either cytosolic or mitochondrially-targeted aequorin using Lipofectamine 3000 reagent (ThermoFisher), as described above. After 24 h of transfection and prior to the experiment, cells were trypsinized, and 4.8 × 106 cells were seeded. At the time of the experiment, the cell medium was removed, and 50 μL of Krebs Ringer modified Buffer (KRB) medium containing 5 μM coelenterazine was added to each well.

Plates were analyzed in a FLUOstar plate reader. Measurements were performed at 1 s intervals for up to 70 s. Five microliters of either 165 μM fluorizoline or 1.1 mM ATP was added after 5 s to achieve a final concentration of 15 μM fluorizoline and 100 μM ATP. At second 35, 5 μL of 12 mM digitonin was added to reach a final concentration of 1 mM digitonin to normalize between experiments and to compensate for differences in transfection efficiency. All the injections present in these experiments were programmed and automatically performed by the fluidics system in the plate reader.

Calcium concentration was calculated as follows:Ratio = (Value − ValueMin)/(ΔValues) ^(1/Slope)^,
[Ca^2+^] = (Ratio + (Ratio · KTR) − 1)/(KR − (Ratio · KR).
where the values for the different constants are:

KR = 10,366,185 M-1; KTR = 120 M-1; Slope = 2.99.

### 4.9. Glutathione Measurement

GSH and GSSG were measured as described by Senft et al. [65]. Cells were collected, washed twice with ice-cold PBS, and then treated with 6% perchloric acid (PCA). Samples were centrifuged at 12,000× *g* at 4 °C for 5 min, and after that, the supernatants were stored at −80 °C. For GSH measurement, samples were incubated with O-Phtalaldehyde (O-PA) for 30 min in the dark. A blank measure was performed by the addition of N-ethylmaleimide (NEM) to the samples. For GSSG measurement, GSH was first sequestered with NEM, and then GSSG was reduced to GSH with Na2S2O3. GSH generated from GSSG was measured by the addition of O-PA. A standard curve with GSH was prepared to extrapolate sample concentration. Samples were loaded into black 96-well Costar plates with flat clear bottom, and fluorescence was measured in a plate reader (ex: 365 ± 5 nm, Emission: 430 ± 20 nm).

### 4.10. Measurement of Oxygen Consumption

High-resolution respirometry was performed using an Oxygraph-2k from Oroboros instruments (Innsbruck, Austria). Jurkat cells were maintained in high glucose RPMI 1640, supplemented as described above. Cells were treated with fluorizoline at 0.7 × 10^6^ cells/mL density, and after 4 h, cells were loaded into 2 mL chambers, and 1 mL was reserved for protein quantification. Routine respiration was measured after its stabilization. After that, 2.5 μM oligomycin and a subsequent titration of carbonylcyanide-4- (trifluoromethoxy)-phenyl-hydrazone (FCCP, 0.5 μM) were added to measure proton leak (LEAK) and maximal electron transfer system capacity (ETS), respectively. Finally, 0.5 μM rotenone and 2.5 μM antimycin A were used to measure residual oxygen consumption, which was subtracted from the previous measurements. All measurements were normalized by protein concentration in the samples, which was measured by BCA.

### 4.11. Statistical Analysis

Statistical analysis was performed with GraphPad Prism 6.0c Software Inc. Results are represented by mean ± standard error of the mean (SEM) of at least three independent experiments. For simple comparisons, a two-tailed Student’s *t*-test was used. For multiple comparisons, a two-tailed ANOVA. *p*-values inferior to 0.05 were considered statistically significant (* *p* < 0.05; ** *p* <0.01; *** *p* < 0.001).

## Figures and Tables

**Figure 1 ijms-22-06117-f001:**
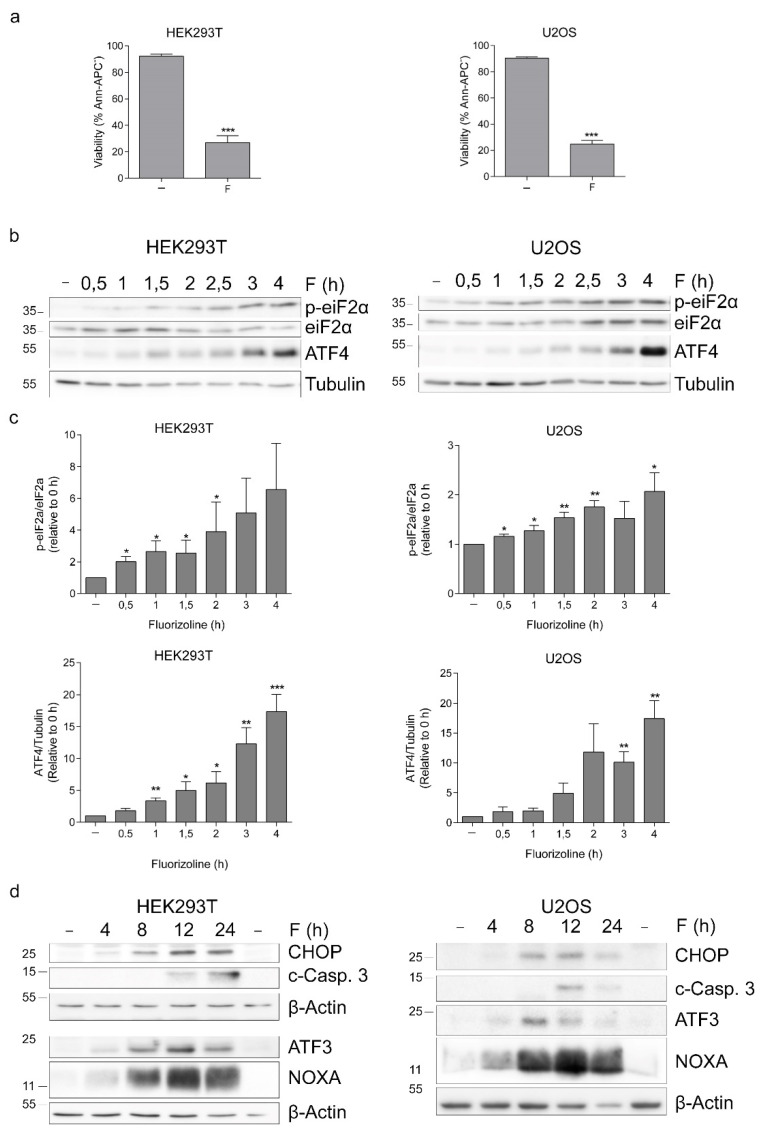
The ISR is activated after fluorizoline treatment in HEK293T and U2OS cells. (**a**) HEK293T and U2OS cells were treated with 15 and 20 μM fluorizoline, respectively, for 24 h. b-d HEK293T and U2OS cells were treated with 15 and 20 μM fluorizoline, respectively, for the indicated time. (**a**) Viability was measured by flow cytometry, and it is expressed as the mean ± SEM of the percentage of non-apoptotic cells (annexin V-negative). (**b**–**d**) Protein levels were analyzed by Western blot. Tubulin and β-actin were used as loading controls. Images proceeding from different gels are separated with white space. These are representative images of at least three independent experiments. (**c**) Quantification of relative p-eIF2α/eIF2 α and ATF4/Tubulin band intensity * *p* < 0.05, ** *p* < 0.01, *** *p* < 0.001. Each condition vs. untreated cells (−) (*n* = 3).

**Figure 2 ijms-22-06117-f002:**
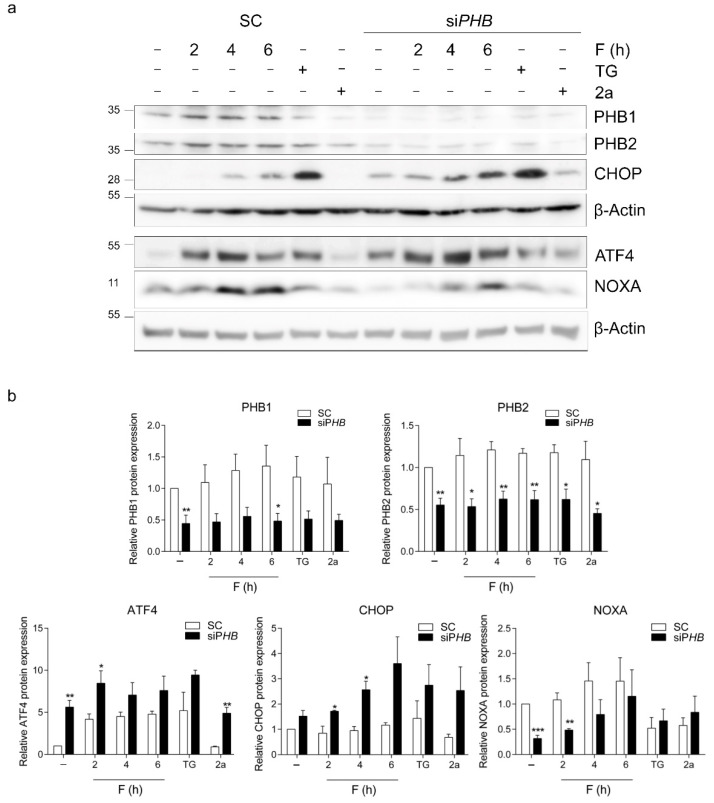
PHB downregulation activates the ISR. HEK293T cells were transfected with scramble (SC) or siRNA against *PHB1* and *PHB2* (si*PHB*) for 72 h. Afterward, cells were treated either with 15 μM fluorizoline (F) for the indicated times, with 20 μM thapsigargin (TG) for 6 h or with 15 μM compound 2a for 6 h. (**a**) Protein levels were analyzed by Western blot. β-actin was used as a loading control. Images proceeding from different gels are separated with white space. These are representative images of at least three independent experiments. (**b**) Quantification of relative band intensity. * *p* < 0.05, ** *p* < 0.01, *** *p* < 0.001. Each condition vs. SC (*n* = 3).

**Figure 3 ijms-22-06117-f003:**
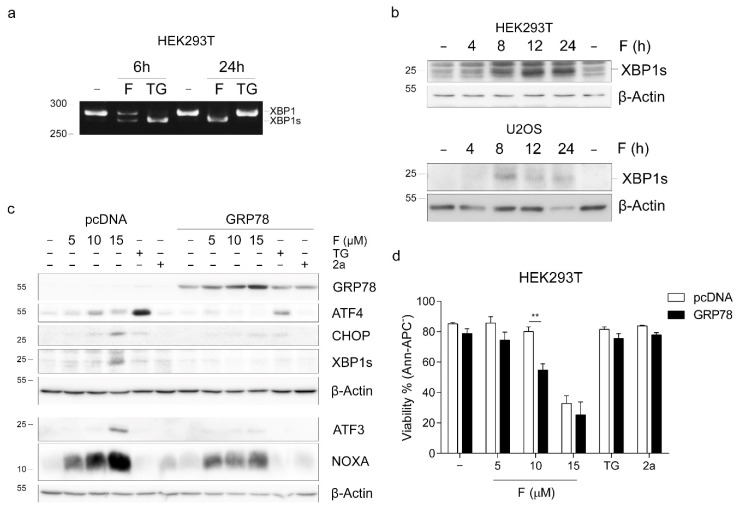
Fluorizoline induces ER stress in HEK293T and U2OS cells. (**a**,**b**) HEK293T and U2OS cells were treated with (**a**) 15 μM fluorizoline (F) or 20 μM thapsigargin (TG) for the indicated time or (**b**) 15 and 20 μM fluorizoline, respectively, for the indicated time. (**c**,**d**) HEK293T cells were transfected with empty pcDNA or with *GRP78* for 48 h and then treated with the indicated doses of fluorizoline, 15 μM compound 2a as a negative control or 20 μM thapsigargin (TG) for 24 h. (**a**) Unspliced (*XBP1*) and spliced *XBP1* (*XBP1*s) mRNA were detected by PCR. This is a representative image of two independent experiments. (**b**,**c**) Protein levels were analyzed by Western blot. β-actin was used as a loading control. Images proceeding from different gels are separated with white space. These are representative images of at least three independent experiments. (**d**) Viability was measured by flow cytometry, and it is expressed as the mean ± SEM of the percentage of non-apoptotic cells (annexin V-negative). ** *p* < 0.01. ISRIB vs. -; GRP78 vs. pcDNA (*n* = 3).

**Figure 4 ijms-22-06117-f004:**
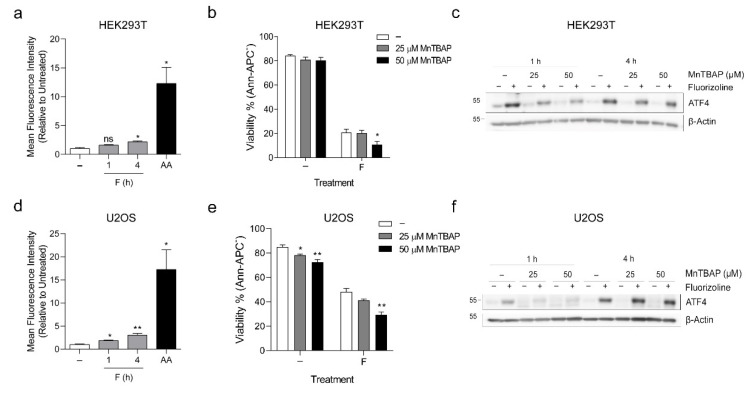
Fluorizoline causes a small increase in ROS production that is not required for apoptosis in HEK293T and U2OS cells. (**a**) HEK293T cells and d U2OS cells were treated with 15 or 20 μM fluorizoline (F), respectively, or 25 and 50 μM antimycin A (AA), respectively, for the indicated time. Cells were loaded with 5 μM MitoSOX 15 min prior to the measurement. Superoxide production was measured by flow cytometry and is expressed as the mean fluorescence intensity. (**b**,**c**) HEK293T and (**e**,**f**) U2OS cells were pre-treated with the indicated doses of MnTBAP for 1 h and then treated with 15 or 20 μM fluorizoline (F), respectively, for (**b**,**e**) 24 h or (**c**), f the indicated time. (**b**,**e**) Viability was measured by flow cytometry, and it is expressed as the mean ± SEM of the percentage of non-apoptotic cells (annexin V-negative). (**c**,**f**) Protein levels were analyzed by Western blot. β-actin was used as a loading control. These are representative images of at least three independent experiments. * *p* < 0.05, ** *p* < 0.01. (**a**) Each condition vs.-; MnTBAP vs.-(**a**,**d** *n* = 4; **b**,**e** ≥ 3).

**Figure 5 ijms-22-06117-f005:**
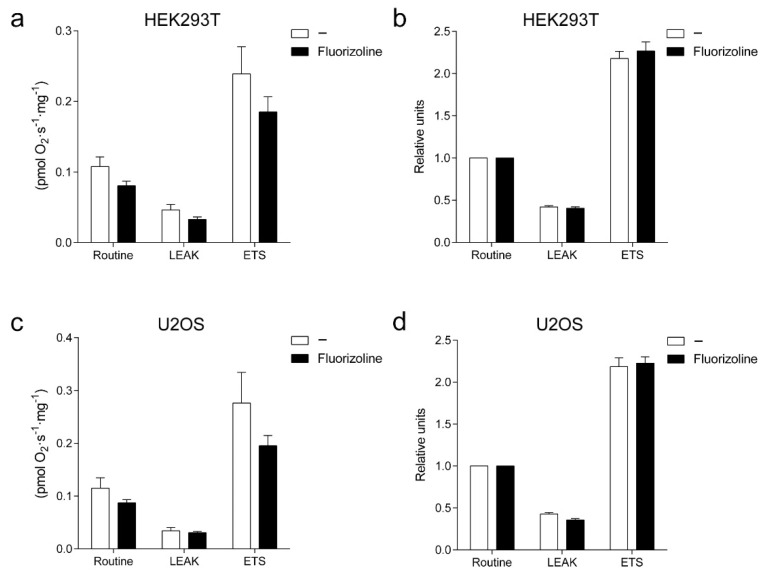
Fluorizoline does not alter mitochondrial respiration in HEK293T and U2OS cells. (**a**,**b**) HEK293T cells and (**c**,**d**) U2OS cells were treated with 15 or 20 μM fluorizoline (F) for 4 h. Oxygen consumption was measured in basal conditions (Routine) after the addition of 1 μM antimycin A (LEAK) and after titration with 0.5 μM CCCP (ETS). Data are represented as (**a**,**c**) raw oxygen consumption rate and (**b**,**d**) oxygen consumption relative to Routine (*n* = 4).

**Figure 6 ijms-22-06117-f006:**
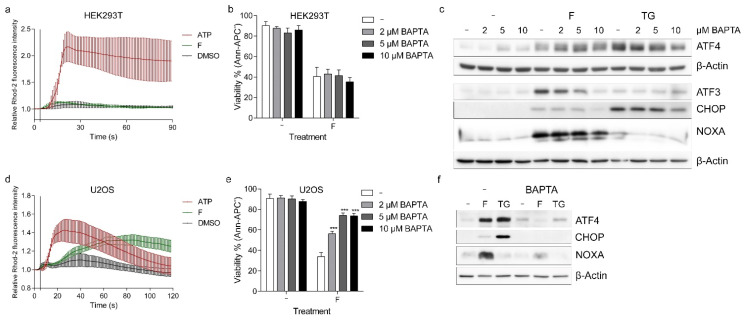
Fluorizoline causes calcium mobilization in U2OS cells. (**a**) HEK293T and d U2OS cells were loaded with 5 μM Rhod-2 30 min prior to the start of the experiment. Live-cell images were acquired at intervals of 1 s for 2 min starting 5 s prior to treatment. HEK293T and U2OS cells were treated either with 100 μM ATP as a positive control, 15 and 20 μM fluorizoline, respectively, or with the corresponding concentration of vehicle (DMSO). (**b**,**c**) HEK293T and (**e**,**f**) U2OS cells were pre-treated with the indicated doses or (**f**) with 10 μM BAPTA for 1 h and then treated either with 15 or 20 μM fluorizoline, respectively, or with 20 μM thapsigargin (TG) for (**c**,**f**) 4 h or (**b**,**e**) 24 h. (**b**,**e**) Viability was measured by flow cytometry, and it is expressed as the mean ± SEM of the percentage of non-apoptotic cells (annexin V-negative). *** *p* < 0.001. BAPTA vs.-(**a** *n* ≥ 3; **b** *n* = 3; **d** *n* = 3; **e** *n* ≥ 3). (**c**,**f**) Protein levels were analyzed by Western blot. β-actin was used as a loading control. These are representative images of at least three independent experiments.

**Figure 7 ijms-22-06117-f007:**
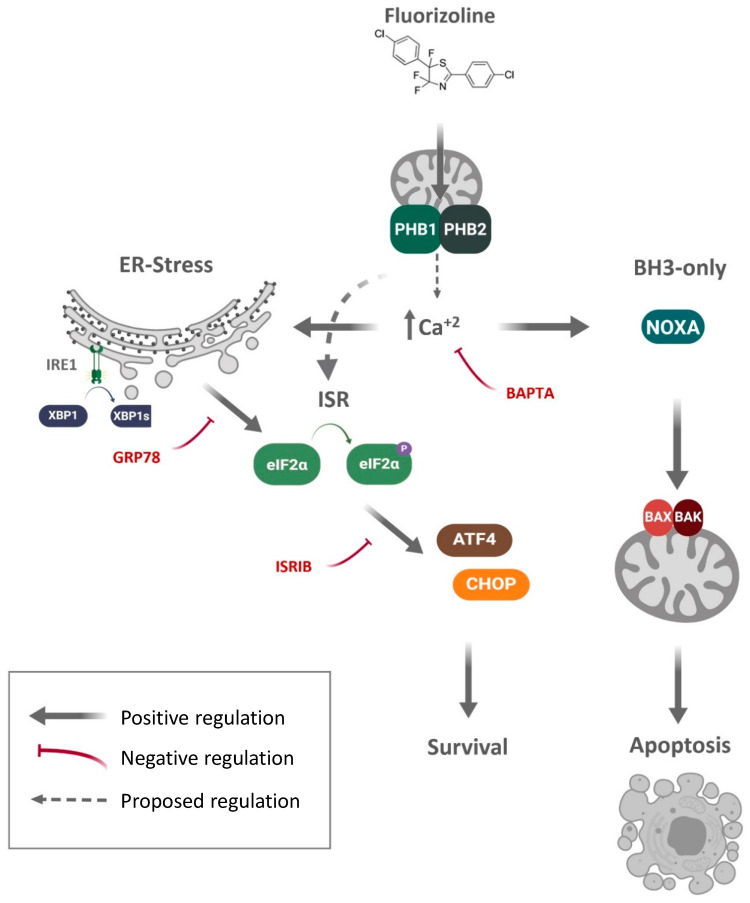
Schematic representation of the pathways involved in the response to fluorizoline in HEK293T and U2OS. The Prohibitin-binding compound fluorizoline triggers the intrinsic pathway of apoptosis through alterations in the expression of BH3-only proteins, such as NOXA. This compound causes increased ER stress leading to the activation of the ISR and increases of ATF4 and CHOP, which have a pro-survival role in the cell types analyzed in the present study. However, this pathway could also explain the modulation of BH3-only protein expression. Fluorizoline can trigger increases in calcium concentration in the cytosol and the mitochondria that are required for the activation of apoptosis, as pre-treatment with calcium chelators, such as BAPTA-AM, can prevent the activation of the ISR, the increase in NOXA expression, and apoptosis in U2OS cells. Grey lines indicate positive regulation; red lines indicate negative regulation. Solid lines indicate described events; dotted lines indicate proposed events.

## Data Availability

Not applicable.

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
