# Peer review of "Activation of the Integrated Stress Response and ER Stress Protect from Fluorizoline-Induced Apoptosis in HEK293T and U2OS Cell Lines"

_ijms, 2021, doi:10.3390/ijms22116117_

Round 1
Reviewer 1 Report
The authors aimed to prove that fluorizoline induces ER/ISR stress response in mainly in HECK293T (human ES embryonic line) and U2OS (human osteosarcoma) cell lines turning to Jurkat (T cell leukemia) cells and this results in induction of apoptosis. I have had many difficulties in following the article due to the numerous testing approaches on different lines making the article very heavy to follow. However the conclusions made on the results do not justify the numerous WB presented. I suggests to re-format the article decreasing the number of cell lines but concentrate on fewer parameters and in some cases better images particularly for NOXA and ubiquitin. Beside ER stress , the authors have not considered the UPR responses.
Reviewer 2 Report
Esteller et al have written a report titled "Activation of the integrated stress response and ER stress protect from fluorizoline-induced apoptosis in HEK293T and U2OS cell lines." The authors show that a compound fluorizoline activates the ISR via calcium release, likely through binding to PHB1/2. While parts of the manuscript are strong, other parts, such as some of the early figures describing depth of the ISR activation are difficult to evaluate (see below). The conclusions are not completely reflected in the results, namely that the ISR is activated through ER stress. Overall, I would recommend major revisions before acceptance.
What does the sentence “Mitochondrial PHB complexes participate in the quality control of nas- 41
cent proteins of the electron transport system, regulate mitochondrial fusion and fission 42
balance as well as cristae morphology, reactive oxygen species formation or mitophagy 43” mean?
It is unclear how PHBs can regulate nascent proteins of the ETS. Do you mean that they regulate mitochondrial protein synthesis?
Page 2, line 64 “with ER stress” not “to ER stress”
Rocaglamide A has been shown to inhibit eIF4A so I am not sure it is the best control if you are tryig to show specificity in PHB/ISR induction. In other words it is a general translation AND ISR inhibitor.
Please replace the NOXA blot in fig2a with a more clear blot. The quantification in fig2b also
does not match the blots in fig2a.
The results in Figs3 are confusing and this may speak to the parts of the apoptotic pathway reported by your methods. Annexin staining is an early phase, and indeed, is variable in cancer cell lines. Caspase cleavage, on the other hand, is very reliable and is considered to report the late stage of apoptosis. Thus I believe caspase cleavage is a better report of what is happening. Indeed in Figs3a you see ATF4 suppresses caspase3 cleavage in 293 cells. CHOP induces caspase 3 cleavage consistent with previous reports (figs3b) if equal loading was achieved (in 293 cells). Finally, the thap control is worrisome. You use at least 10-times the concentration most people use in these cell lines, and for 6x the amount of time for only a moderate effect. I wonder if this is also confounding your data. There should be no cells still attached under these conditions if thapsigargin were working normally. That said annexin does seem to follow caspase 3 in U2OS cells. So I am left unsure of the conclusions and point of using both cell lines.
You say “The addition of ISRIB reduced the increase of ATF4 and CHOP after 4 190
hours of treatment with fluorizoline, indicating that the induction of these two proteins is 191
caused by the phosphorylation of eIF2(Figure S4b).”. However, you did not measure eIF2a so how do you know? Additionally, there is only a modest change in ATF4 and CHOP, if any. At the very least you should provide a blot for eIF2a as well as quantification of ATF4 and CHOP. In general, the data does not support the conclusions you make about the ISR because rarely is P-eIF2a tested and activation of the kinases is not evaluated.
In general, the blot quality in the first figures is poor. Between variable protein loading (probably often because the cells are sloughing off the plate and/or dead at the time of harvest), or having nice crisp bands. This makes it difficult to evaluate what is going on. On top of that the quantification is often different or there are inconsistencies between cell lines or methods for measurement (annexin versus caspase cleavage). That said the quality does appear to improve fig4 on.
You say there is overexpression of GRP78 in figs5 but do not show it. How is it possible to evaluate the results given the inconsistency between the annexin and caspase3 cleavage and not showing GRP78 overexpression?
Page 9 line 249 ubiquitin is misspelled
Overall the analysis of calcium effects are strong. I would request that you explain the use of digitonin in the results when you explain aquorin results. Would also request that you add P-eIF2a lots throughout especially in figure 6.
Also, can you perform qPCR for some of the ATF4 targets in these cells under fluorozine treatment?
In the discussion you indicate that the effects of ISRIB on CHOP suggest a translation effect. You should back this up with more data as Figs4 is not convincing. You could use a CHOP reporter as has been previously published by David Ron and by blotting for ATF6. It would also be useful to show P-PERK levels under fluorizoline since you strongly argue throughout the ISR is activated through ER stress.
Page 17 line 480 you say “independent on” where it should read “independent of”.
Please better explain the discrepancy between the cell lines in the discussion. In other words, why might there be different sensitivities. I realize you may not know but there should be some testable hypotheses.
Reviewer 3 Report
The authors have addressed the majority of the concerns
However, in the new figure S2, the level of Noxa is very high in basal condition, in both cell lines (which is not the case in all the others figures of the paper). Why? There is also some experimental explanations missing in the corresponding figure legend
Author Response
Following the suggestion by reviewer 3 we have replaced the images in Figure S2a by new blots. We think that the discrepancies with other figures in the article can be caused by a longer exposition and a darker presentation. In the case of Figure S2b the images have been maintained but brightness has been corrected to be more homogeneous with other blots in the article. We have also corrected the description in the corresponding figure legend.
Reviewer 4 Report
Authors have addressed my comments.
Author Response
We appreciate very much the work of Reviewer 4 in contributing to improve the quality of the manuscript.
Round 2
Reviewer 1 Report
I still have some concern regarding the WB results, in particular NOXA expression. How did the authors manage to quantify such level of expression?
Reviewer 2 Report
I apologize for the confusion with respect to your manuscript review and processing. AT this point I would recommend the manuscript for acceptance based on your responses and that fact that it has already been reviewed thoroughly by other reviewers. In addition to your response to those reviewers, you have supplied satisfactory responses to my inquiries.
For the record, most publications describing annexing staining indicate it is an early stage of apoptosis. Some cancer cell lines show high basal staining and others never show a robust increase despite caspase 3 activation. My main point here was that I was very srurprised under your control thapsigargin condition there were any cells remaining to use for western blot given the conditions of the assay.
As for ISRIB, some have shown there is a difference in P-2a while others have shown the opposite. Which is it in your cells/conditions?
These questions/comments are for consideration in your future work. Thank you for taking the time to respond. Best of luck.
This manuscript is a resubmission of an earlier submission. The following is a list of the peer review reports and author responses from that submission.
Round 1
Reviewer 1 Report
The study conducted by Saura-Esteller and collaborators is focused on the activation by a PHB-interacting drug, fluorizoline, of the integrated stress response and ER stress in human cell lines. This team is implicated for a long time in the molecular way of action of fluorizoline. Although the data presented are convincing, the global organization may be modified to show in the main figures the results obtained that are common in all the cell lines tested to define more clearly the general fluorizoline action on ISR and ER stress in all, and put in supplementary figures what is cell type specific.
There is other fluorizoline targeting compound, such as Rocaglamide. What is the effect of RocA on ISR and ER stress activation? As Rocaglamide is not inducing apoptosis, this may help to define if the activation of these pathways are linked or not.
In figure 2, the authors show that PHB silencing activates the ISR. However, in panel a it seems that there is an additive effect of fluorizoline treatment in PHB silenced cells, suggesting that fluorizoline activation of ISR could be independent of PHB. Can the authors comment on this point? Can fluorizoline have other target?
Line 160, the authors wrote that “the increase of ATF3 protein levels was not prevented by ISRIB indicating that this induction is not controlled by ATF4”. As ATF4 expression is still present in fluorizoline-ISRIB treated cells, the authors may modifcy their conclusion. What can be the role of ATF3 induction by fluorizoline?
Data presented in figure 5 may also be done on 293 and U2OS cells, to have consistency in the cell line used in the main figures, and in order to draw a general mode of action of fluorizoline (referred to first reviewer comment).
Mino points:
Improve the quality of the MTDR pictures in figure S4
Reviewer 2 Report
The authors examined the mechanism by which fluorizoline, a prohibitin-binding compound, induces/regulates cell death in HEK293T and U2OS cells. In HEK293T cells, fluorizoline-induced apoptosis was enhanced by overexpressed GRP78, ATF4 silencing, and the integrated stress response (ISR) inhibitor ISRIB (although the increase in apoptosis was only marginal, as pointed out below), suggesting that the ER stress response/ISR protect HEK293T cells from fluorizoline-induced apoptosis. On the other hand, CHOP, but not ATF4, was suggested to protect from fluorizoline-induced apoptosis in U2OS cells. Induction of apoptosis by fluorizoline depended on intracellular calcium in U2OS cells, but not in In HEK293T cells. The Jurkat cell line was used to test whether fluorizoline causes ER stress by altering ROS production, GSH concentration, or mitochondrial respiration. Fluorizoline did not alter them in Jurkat cells.
The authors have previously demonstrated that ER stress is involved in fluorizoline-induced apoptosis in Hela cells and HAP1 cells. Taking this previous study and the current study together, it appears that the mechanism of cell death induced by fluorizoline is highly dependent on the cell type. Therefore, it may be inappropriate to describe the inhibition of fluorizoline-induced apoptosis by ISR/ER stress as a general phenomenon (especially in the Title, Abstract, and Fig. 6), and it should be emphasized that it may occur in limited situations, such as in HEK293T and U2OS cells. In addition, it is not explained why ATF4 and CHOP exert cell type-specific protective effects against fluorizoline-induced apoptosis in HEK293T cells and U2OS cells, respectively. However, more problematic is that the data are not clear, and authors need to interpret their experimental results more carefully and/or show additional evidence to strengthen their conclusion.
What is the significance of elucidating the mechanism of fluorizoline-induced cell death? Is this compound a promising candidate for an anticancer drug? Has it been reported that it can treat cancer in vivo?
siATF4 and ISRIB only marginally enhanced apoptosis in HEK293 cells incubated with 10 μM of fluorizoline (Fig. 3b and Fig. S2a). Moreover, overexpression of GRP78 enhanced apoptosis when HEK293T cells were treated with fluorizoline at a concentration of 10 μM, but not 5 μM or 15 μM (Fig. 4d), indicating that the protective effect of ISR/ER stress against fluorizoline-induced apoptosis is seen only in limited settings (cell type and dose). Hence, the present set of data are not sufficiently convincing to draw the conclusion that ISR and ER stress protect from fluorizoline-induced apoptosis. The authors may test whether induction of active caspase-3, caspase-3 activity, and PARP cleavage by fluorizoline are increased by siATF4, siCHOP, ISRIB, and GRP78 in parallel with the induction of apoptosis.
Mitochondrial respiration and ROS production were assessed only in Jurkat cells (Fig. 5). They should perform the same experiments with HEK293T and U2OS cells, as the main focus of this study is on these cell lines.
BAPTA-AM significantly reduced fluorizoline-induced apoptosis in U2OS cells (Fig. 6e). However, EGTA-AM did not or only marginally reduced it (Fig. S6). Is it possible that these contradictory results are due to off-target effects of BAPTA-AM?